# Characterization of Microstructure, Phase Composition, and Mechanical Behavior of Ballistic Steels

**DOI:** 10.3390/ma15062204

**Published:** 2022-03-16

**Authors:** Waseem Khan, Muhammad Tufail, Ali Dad Chandio

**Affiliations:** Materials and Surface Engineering Laboratory, Department of Metallurgical Engineering, NED University of Engineering and Technology, Karachi 75270, Pakistan; waseemkhan@neduet.edu.pk (W.K.); pvc@neduet.edu.pk (M.T.)

**Keywords:** ballistic steels, variable alloy thicknesses, microstructures, mechanical properties, fractography, corrosion resistance

## Abstract

For the protection of civil and military armored vehicles, advanced steels are used, due to their outstanding mechanical properties, high ballistic performance, ease of manufacturing and low cost. However, after retrofitting, weight is the prominent issue. In this regard, several strategies are being proposed, which include the surface engineering of either low-thickness ballistic steels or conventional steels, in addition to new alloys and composites. Therefore, to better understand the response of such materials under various stimuli, the existing state of the art ballistic steels was utilized in this study. The aim of this study was to better understand the existing materials and their corrosion behavior. Therefore, in this connection, two thicknesses were selected, i.e., thin (6.7–7.0 mm) and thick (13.0–15.0 mm), henceforth termed as low thickness (LT) and high thickness (HT), respectively. This was followed by characterization using tensile, Charpy, micro-Vickers, nanoindentation, XRD, SEM-EDS and corrosion tests. Microstructurally, the LT samples only exhibited ε-carbide precipitates, whereas the HT samples contained both ε-carbides and Mo_2_C (molybdenum carbides). However, both samples were found to be tempered martensite with a lath morphology. Moreover, higher hardness, and lower elastic modulus and stiffness were noticed in the HT samples compared with their LT counterparts. Fractured surfaces of both of these alloys were also examined, wherein a ductile mode of fracturing was observed. Further, a corrosion study was also carried out in brine solution. The results showed a higher corrosion rate in the HT samples than that of their LT counterparts. An extensive discussion is presented in light of the observed findings.

## 1. Introduction

Modern small range arms cause an extreme threat to both civilian and military vehicles and infrastructure worldwide. However, protective materials used in such vehicles play an important role in minimizing or preventing the associated threats. Thus, materials such as steels [1], composites and ceramics [2] have earned significant importance in providing the desired protection. The performance of materials is largely dependent on the processing, structure and property co-relationships. Among all metallic armor materials, high-strength ballistic steels are widely used to avert threats at different levels. In addition, they offer the combination of high strength, hardness and toughness, excellent ballistic performance, good machinability, ease of heat treatment and weldability [3].

Ballistic steels are made into slabs by the casting process, and then hot rolled into plates of the desired thicknesses. This is also known as rolled homogenous armor (RHA) steel [4]. Next, they are subjected to austenitizing, quenching and tempering processes to obtain the desired properties [5]. Thus, in the development of high-strength ballistic steel plates, alloying, rolling and heat treatment processes play a vital role [6,7]. Moreover, the ballistic performance of steels (e.g., 50Cr4V steel) depends on the hardness, toughness and strength. Thus, improvements in such properties will depend on the microstructure, e.g., tempered martensite, etc., in addition to other factors. This necessitates an optimal combination of strength, hardness and toughness [8,9,10,11]. Material properties versus projectile impact has been of practical significance for ballistic protection applications [12], in addition to the thickness of steel plates [13,14].

The microstructure of ballistic steels usually consists of martensite, bainite or tempered martensite, or a combination thereof [15]. Specifically, the martensitic phase provides high hardness and strength levels in steel. It should be noted that the strength and hardness are increased by the quenching process, whereas tempering increases the ductility and toughness in steels, thus enhancing the impact energy absorption capability [16,17,18].

Moreover, it is of paramount importance to predict the ballistic performance of steels, since mechanical properties alone do not suffice. Thus, the microstructural approach is equally important in this regard [19,20,21,22]. Processing parameters play an important role too, such as tempering temperature, etc. For example, it has been reported that the mechanical properties of high-strength ballistic steel are significantly enhanced at lower tempering temperatures [23]. Microstructurally, a combination of martensite and bainite phases is instrumental for the desired performance of steels [24]. In addition, even the morphologies of martensite must be examined, such as the butterfly, lath, lenticular and plate types, since such morphologies are responsible for the desired properties of steels. Lenticular martensite forms in steel with a high carbon content, whereas lath martensite forms in steel with a low carbon content [25]. Lath morphology in a tempered martensite structure increases the dynamic strength of steel [26]. Further studies on steels showed that alloying elements such as carbon, manganese, boron, nickel and molybdenum improves their mechanical characteristics [27], since alloyed elements are important constituents in the formation of various precipitates, such as carbides, that are beneficial for the strength of steels [28].

Moreover, corrosion resistance is another property that is as important as the desired protection. The literature suggests that martensite exhibits lower corrosion resistance than ferrite and other phases, since corrosion depends on the nature, size and distribution of phases in the microstructure [29,30].

Therefore, in the present study, ballistic steel plates of various thicknesses were studied to better understand their microstructure, mechanical properties and corrosion resistance, since, to the best of the authors’ knowledge, there are no corrosion studies that have been conducted on ballistic steels in brine solution. In addition, the focus of this study is on LT steels and how they differ from their HT counterparts. This is because the weight of the vehicle after armoring is an issue of concern, since suspension, transmission and other parts (of the vehicle) need improvement. This will facilitate the search for an alternative material with a similar microstructure to the HT counterpart.

## 2. Experimental Work

### 2.1. Materials

The main material for this study is the state-of-the-art ballistic steels (Pak Armoring & Streit Pakistan, Karachi, Pakistan), which are currently being used for both military and civilian vehicles. The steel plates were obtained from local markets, as shown in Table 1.

Moreover, the chemical composition and mechanical properties of standard material are shown in Table 2 and Table 3, respectively [31,32].

### 2.2. Methods

#### 2.2.1. Metallography

The ballistic steel samples were cut into 15 mm × 15 mm squares using an abrasive cut-off machine (Metkon, Model: METACUT M-250, Bursa, Turkey), and cold mounted in epoxy resin (Allied High Tech Products Inc., Compton, CA, USA). This was followed by grinding progressively from 240 to 1200 grit silicon carbide papers. Thereafter, polishing was carried out with different sizes of diamond paste, i.e., 5 μm, 3 μm and 0.5 μm, using a polishing machine.

#### 2.2.2. Etching

Samples were etched after polishing using 2% Nital (2 mL HNO_3_ and 98 mL methyl alcohol) (Sigma-Aldrich, St. Louis, MI, USA) at room temperature to reveal their microstructures.

### 2.3. Testing and Characterization

#### 2.3.1. Optical Emission Spectroscopy (OES)

Samples were cut into 30 mm × 30 mm sizes using the abrasive cut-off machine to find out the chemical composition. Optical emission spectroscopy (Ametek, Model: SPECTROMAXx, Berwyn, PA, USA) was used to measure the chemical composition of samples.

#### 2.3.2. Microscopy

The microstructures of the ballistic steel samples were obtained using an optical microscope (Metkon, IMM-901, Bursa, Turkey) and a scanning electron microscope (Tescan, Series: VEGA3, Brno, Czech Republic). For microstructural details, the scanning electron microscope was operated at 20 kV and at different magnifications. An energy-dispersive X-ray spectrometer (Brucker, Model: 1119-400, Billerica, MA, USA) was used with SEM for further chemical composition analysis of ballistic steels.

#### 2.3.3. X-ray Diffraction

For X-ray diffraction (XRD) measurement, rectangular samples of 2 mm thickness with a dimension of 40 mm × 20 mm were cut from each of the plates using a wire cutting machine. The samples were ground from 320 to 1000 grit silicon carbide paper and then polished using 5 μm diamond paste. The samples were then ultrasonically cleaned in an ethanol bath. An X-ray diffractometer (PANalytical, Model: PW3040/60 X’Pert Pro, Amsterdam, The Netherlands) was used to analyze different phases in the samples. The X-ray diffractometer was run at a current of 30 mA and a voltage of 40 kV, and scanned over 1 h, at 2θ angles from 10 to 80° with a step of 0.025° and CuKα radiation, at room temperature. The acquired XRD peaks were examined by the comparison method with the help of the JCPDS data file 65-4899 [33].

#### 2.3.4. Tensile Testing

Tensile testing procedures as well as the geometry of samples are prescribed in the ASTM E8 standard. The samples were run at a speed of 5 mm/min using a hydraulic universal testing machine (Daekyung Tech & Testers, Model: DTU-900HCB, Incheon, Korea) at room temperature, to determine their tensile properties. The yield strength of samples was determined at 0.2% strain using the offset method. Three samples from each steel plate were tensile tested and the average value for each steel plate was measured.

#### 2.3.5. Charpy Impact Testing

For preparation of standard Charpy, V-notch samples were machined in a size of 10 mm × 10 mm × 55 mm for larger thickness plates and in a size of 5 mm × 10 mm × 55 mm (subsize sample) for smaller thickness plates, with a 2 mm depth notch, using the wire cutting machine as prescribed in the ASTM E23 standard. Impact testing of samples was carried out by using the pendulum impact testing machine (Wance, Model: 452G-3, Shenzhen, China). During impact testing, a hammer of 300 joules was used. Three samples from each steel plate were impact tested and the average value of each sample was measured.

#### 2.3.6. Vickers Microhardness Testing

The samples for Vickers microhardness testing were prepared according to the methods defined in Section 2.2.1 and Section 2.2.2. The hardness of samples was determined as per ASTM E92 using Vickers microhardness tester (Shimadzu, Model: HMV-G31, Kyoto, Japan) under applied load of 500 g and holding time of 10 s. Ten indents were recorded at different locations and their average values are reported. The bulk hardness was determined.

#### 2.3.7. Nanoindentation Testing

In addition to Vickers microhardness, nanoindentation testing was also carried out as per the procedure defined in Section 2.2.1 and Section 2.2.2. Nanoindentation of samples was performed according to ASTM E2546 using a nanoindenter (Anton Paar, Model: TTX-NHT^3^, Graz, Austria).

A diamond Berkovich indenter (Anton Paar, TTX-NHT3, Graz, Austria) was used by applying a load of 100 mN for a holding time of 15 s. A microscope at 1000× magnification was attached to the nanoindenter that was used to reveal various phases in the microstructures of samples. During the nanoindentation test, the loading and unloading rate was 200 mN/min. To determine the micro-mechanical properties, such as indentation hardness, elastic modulus and stiffness, an array of three nanoindentations were performed at different locations, and the average values were calculated using the Oliver–Pharr method [34]. To avoid possible effects of plastic zone overlapping in samples, spacing of 4 μm between the two indents was maintained [35].

#### 2.3.8. Fractography

After mechanical testing, the fracture analysis of tensile and Charpy impact samples was carried out using a scanning electron microscope at different magnifications.

#### 2.3.9. Corrosion Testing

Samples for corrosion testing were prepared using a single copper wire soldered with steel and cold mounted in epoxy resin, leaving an area of 0.2 cm^2^ exposed to the electrolyte. This was followed by the grinding and polishing of samples using the aforementioned metallographic procedure. Corrosion testing of samples was carried out according to the ASTM G59 standard. Corrosion behaviors of all the samples were determined by the potentiodynamic polarization method in 3.5 wt.% NaCl solution at room temperature using a potentiostat (Gamry Instruments, Model: Reference 600, Pennsylvania, United States of America). An electrochemical cell was used, which consisted of a three-electrode system—a reference electrode (Ag/AgCl), a counter electrode (graphite) and a working electrode (ballistic steel samples). After attaining equilibrium open circuit potential, polarization potential was applied from −2.5 V to 2.5 V at a scan rate of 0.5 mV/s to evaluate the corrosion resistance of the samples. Three tests were performed for each sample to assess their behavior.

## 3. Results and Discussion

### 3.1. Chemical Composition

The chemical composition of the samples is presented in Table 4, using optical emission spectroscopy (OES). Ballistic steels consist of low carbon, Mn, Si, Cr, Ni and Mo contents. The LT samples contained higher chromium and nickel contents in comparison to the HT samples. The measured chemical composition of the samples is nearly the same as that of the standard material (Armox 500T), as shown earlier in Table 2.

### 3.2. Microstructure

#### 3.2.1. Optical Microscope

The microstructures of the ballistic steel samples were analyzed using an optical microscope at 1000× magnification, as shown in Figure 1. The microstructures were revealed to be tempered martensite, and precipitation of carbides was also noticed in all the samples. However, martensite was more homogenously distributed in the HT samples than in the other counterparts. This facilitates increased strength, hardness and toughness. In general, a lath martensite morphology can be observed in all the microstructures.

#### 3.2.2. Scanning Electron Microscope

Scanning electron microscopy was used to analyze the microstructures of the ballistic steel samples at varying magnifications. The SEM micrographs are shown in Figure 2a–d. The presence of martensite laths with carbide precipitations was also observed, which is consistent with the optical images. In particular, the lath and grain sizes of martensite are nearly similar in all the samples. However, no retained austenite was observed in the samples, since it was transformed into ferrite and carbides. In the HT samples, epsilon (ε) carbides and molybdenum carbides were observed, while LT only exhibited ε-carbides.

Moreover, the EDS point analysis of the LT samples only indicated the precipitation of ε-carbides (see Figure 3), while the HT samples exhibited (see Figure 4) the precipitation of both ε-carbides and Mo_2_C (molybdenum carbides), which is consistent with previous studies [36,37].

### 3.3. X-ray Diffraction Analysis

The X-ray diffraction patterns of the ballistic steel samples are shown in Figure 5. The X-ray diffraction patterns only show martensitic peaks, i.e., (110)_α’_ and (200)_α’_, in both the LT and HT samples, respectively. It should be noted that no retained austenite was observed in the XRD analysis, which further confirms the fully tempered martensite structure in all the samples.

### 3.4. Mechanical Properties

#### 3.4.1. Tensile Properties

The tensile properties, and some other properties, such as toughness and hardness, were also measured for the ballistic steel samples, as shown in Table 5. Figure 6 shows engineering stress and strain curves of the ballistic steel samples, whose detailed extracted data are shown in Figure 7. It can be analyzed from the data that the yield, elongation and tensile strength values of the HT samples are higher than those of their LT counterparts, due to a scale effect. Percent elongation represents the ductility of the material, thereby suggesting that the HT samples are more ductile than their LT counterparts. This change in properties is attributed to the microstructure, i.e., precipitates and homogeneity. In addition, it is due to a slight variation in the chemical composition of both sets of samples, as shown in Table 4.

#### 3.4.2. Impact Resistance

The impact properties of the ballistic steel samples were also measured using the Charpy test. Impact energy is an indication of the energy required for crack initiation and propagation. Thus, the toughness values for the LT samples were measured to be 38 and 44 joules for LT-1 and LT-2, respectively. On the contrary, 68 and 55 joules were obtained for HT-1 and HT-2, respectively, as represented in Table 5 and Figure 8. This suggests that the LT-2 sample is tougher than the LT-1 sample in the same category, while HT-1 appeared to be tougher than its HT-2 counterpart. However, in general, the HT samples were revealed to be tougher than their LT counterparts.

#### 3.4.3. Vickers Microhardness

The hardness of the ballistic steel samples was measured using Vickers microhardness tester. The hardness values for the LT samples were measured to be 535 VHN (Vickers hardness number) and 560 VHN for LT-1 and LT-2, respectively. The HT samples were measured to be 590 VHN and 550 VHN for HT-1 and HT-2, respectively, as shown in Table 5 and Figure 9. The hardness values of the samples indicate that the LT-1 sample is harder than the LT-2 sample, whereas the HT-1 sample is harder than the HT-2 sample. However, in general, the results revealed that the HT samples are harder in comparison with the LT samples.

#### 3.4.4. Nanoindentation Analysis

To better understand the mechanical properties of ballistic steel, such as the indentation hardness (H), elastic modulus (E), reduced elastic modulus (E_r_) and stiffness (S), nanoindentations were carried out. Thus, the indentation hardness (H) was calculated by using Equation (1) [34,38]:(1)$$H=\frac{F_{\max}}{A_{P}}$$
where F_max_ = maximum load and A_p_ = indentation projected contact area.

Moreover, the reduced elastic modulus (E_r_) was calculated by using Equation (2) [34,38]:(2)$$E_{r}=\frac{\sqrt{\pi }\cdot S}{2\cdot \beta \cdot \sqrt{A_{P}\left(h_{c}\right)}}$$
where S = stiffness, A_p_ = indentation projected contact area, h_c_ = indentation contact depth and β = geometric constant with a unity value.

In addition, the plain strain elastic modulus (E^*^) was determined by using Equation (3):(3)$$E^{*}=\frac{1}{\frac{1}{E_{r}}-\frac{1-\nu_{i}^{2}}{E_{i}}}$$
where E_i_ = elastic modulus of the indenter and ν_i_ = Poisson’s ratio of the indenter.

The elastic modulus (E) was calculated by using Equation (4) [34,38]:(4)$$E={\;E}^{*}\cdot \;\left(1-\nu_{s}^{2}\right)$$
where E = elastic modulus of the sample and ν_s_ = Poisson’s ratio of the sample.

The load–displacement curves of the ballistic steel samples obtained using the nanoindentation tester are shown in Figure 10. In addition, the micromechanical properties, such as the indentation hardness, elastic modulus and stiffness, are shown in Table 6. The HT samples (in particular, the 15 mm sample) exhibited higher indentation hardness, and lower elastic modulus and stiffness than the other counterparts. This is due to the homogeneous distribution of lath martensite and carbide precipitation. This further suggests that for ballistic protection applications, the martensitic phase is the most desirable structure of steels. This is because it provides optimum mechanical and ballistic performance [23].

### 3.5. Fractography 

#### 3.5.1. Tensile Samples

A scanning electron microscope was used to examine the fractured surfaces of the samples, so as to understand the mode of failure. Figure 11 illustrates fractured surfaces of tensile samples, whereby necking regions could be observed in the samples. Fibrous and shiny surface features, along with significant amounts of deformation, thereby forming shearing lips at the corners of the samples, could be observed. Therefore, failure occurs after necking in tensile samples by localized shear plastic. It should be noted that the dimple is a distinguished feature of a ductile fracture, as observed in the present case.

Moreover, micro-voids and smooth facets can be observed in all the tensile samples, since fracturing takes place in alloys by micro-void nucleation and coalescence. This is due to a pile-up of dislocations, grain boundaries, inclusions and carbide particles [39]. Figure 11 represents large amounts of variation in dimple size and shape on the fracture surfaces. Thus, all the samples exhibited ductile failures.

#### 3.5.2. Impact Samples

Figure 12 shows fractured surfaces of the samples after impact testing. All the fractured surfaces exhibit equiaxed dimples, along with micro-voids, similarly, presumably, to the crack initiated and propagated from the region of micro-voids [39]. The similar morphology of the dimples in all the samples suggests that this is a ductile failure. This is consistent with the observations of tensile fractures discussed earlier.

Moreover, it should be noted that hardness, strength, and toughness are properties of prime importance for ballistic steels, which were found to be better for the HT samples than for all the other samples presently studied.

#### 3.5.3. Microstructure and Mechanical Behavior Co-Relationships

The morphology of the fracture surfaces of the samples after both the tensile and impact tests shows a relation between the mechanical behavior of the samples and the microstructure of the corresponding materials. As mentioned above, all the microstructures were characterized by a lath martensite morphology. However, a great degree of homogeneity of tempered martensite was observed in the HT samples compared to that of their LT counterparts. In the LT samples (Figure 1a,b and Figure 2a,b), quite distinct packets of martensite laths can be observed. The average size of the packets is about 7–10 μm. For the HT-2 sample, the packets of martensite laths are about the same size, but their contours are blurred (Figure 1d). In contrast to this, a homogeneous lath martensite microstructure is observed for the HT-1 sample (Figure 1c and Figure 2c). As can be observed in Figure 11c and especially, in Figure 12c, the fracture surface of the HT-1 sample contains plastic elongation crests surrounding a set of dimples. Such crests are of a greater size (about 50–100 μm, see Figure 12c), and are more distinct than those of other variants. That is why, at similar levels of tensile strength, greater elongation is achieved for the HT-1 sample (see Table 5). Such a ductile fracture micromechanism also explains the increased toughness (impact resistance), due to enhanced ductility of the material (see Table 5).

### 3.6. Corrosion Analysis

Potentiodynamic polarization curves were obtained in 3.5 wt.% NaCl solution, as shown in Figure 13. The corrosion current density (I_corr_) and corrosion potential (E_corr_) were determined by Tafel extrapolation techniques, whereas the corrosion rate was calculated using Equation (4) [40].
(5)$$\mathrm{Corrosion}\;\mathrm{rate}\;\left(\frac{\mathrm{mm}}{\mathrm{year}}\right)=\frac{I_{\mathrm{corr}\;}\times 0.00327\times \;\mathrm{EW}}{\rho }$$
where $I_{\mathrm{corr}}$ = corrosion current density (µA/cm^2^), EW = equivalent weight of iron (27.92) and ρ = density of ballistic steel (7.850 g/cm^3^).

The corrosion parameters were obtained from Tafel curves using the extrapolation technique, as shown in Table 7. In general, the results revealed that the LT samples exhibited a lower corrosion current density (I_corr_), corrosion potential (E_corr_) and, hence, corrosion rate. On the other hand, the HT samples exhibited a higher corrosion current density (I_corr_), corrosion potential (E_corr_) and, thus, corrosion rate. The reduction in the corrosion rate of the LT samples is due to the presence of ε-carbides and more chromium, and the less homogeneous distribution of the martensite phase. It should be noted that the tempered martensite structure always exhibits a larger corrosion rate when compared to other steels. Nevertheless, the increase in the corrosion rate of the HT samples is attributed to the more homogeneous distribution of the martensite phase [40], ε-carbides [41] and Mo_2_C-carbides [42]. Presumably, this provides more nucleation sites for corrosion cells to develop. Thus, the LT samples are more corrosion resistant when compared to their HT counterparts.

## 4. Conclusions

Based on the present set of experimental conditions adopted for low-thickness and high-thickness ballistic steel samples, the following are the concluding remarks. The aim of this study was to better understand the existing materials used for armoring applications, in particular, for civil and military vehicles, in addition to their corrosion behaviors.

The microstructures of all the samples were analyzed, wherein the LT samples only exhibited ε-carbide precipitates, whereas the HT samples contained both ε-carbides and Mo_2_C-carbides. In addition, a greater degree of homogeneity of tempered martensite was observed in the HT samples than in their LT counterparts. Hence, improved mechanical properties (i.e., tensile properties, toughness and hardness) of the same were observed. In addition, the fracture modes were ductile in nature, which implies ease of application of such materials. Moreover, the corrosion resistance of the HT samples was found to be poor in comparison to their LT counterparts. This was attributed to the microstructure, i.e., homogeneity and precipitates. Thus, these findings will facilitate the development of conventional steels for potential armoring applications. Therefore, the HT samples are a better choice for armoring applications at an advanced level of protection.

## Figures and Tables

**Figure 1 materials-15-02204-f001:**
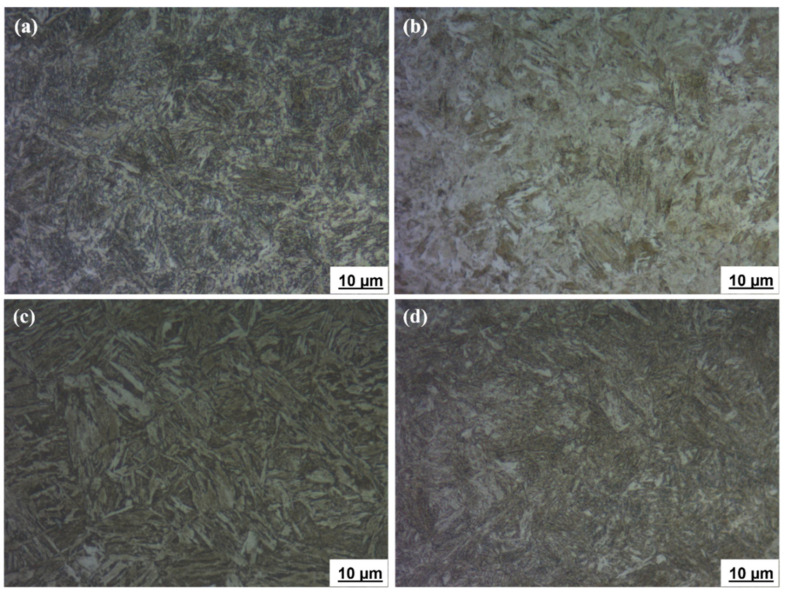
Optical microstructures of ballistic steel samples: (**a**) LT-1; (**b**) LT-2; (**c**) HT-1; (**d**) HT-2.

**Figure 2 materials-15-02204-f002:**
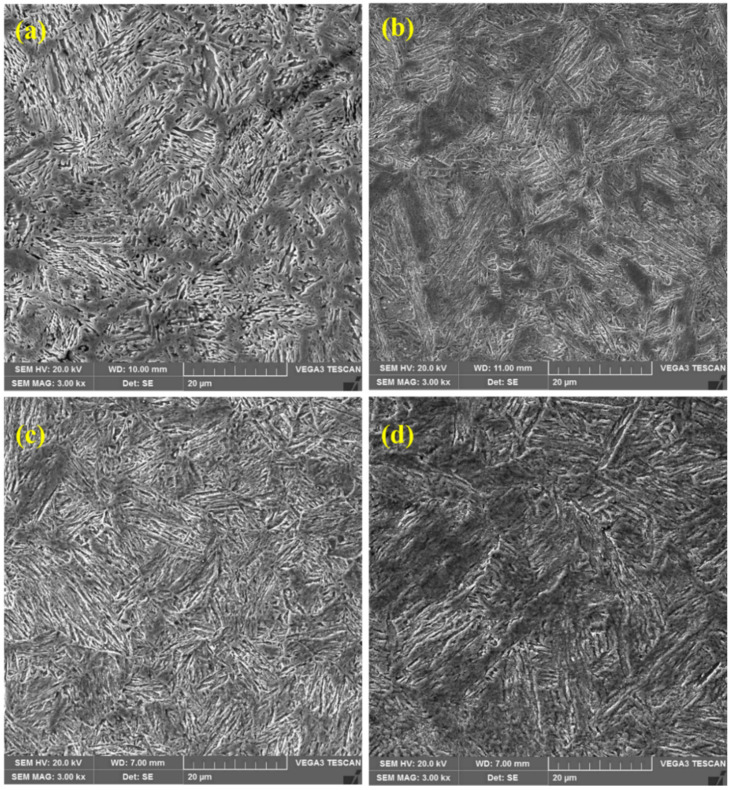
SEM micrographs of ballistic steel samples: (**a**) LT-1; (**b**) LT-2; (**c**) HT-1; (**d**) HT-2.

**Figure 3 materials-15-02204-f003:**
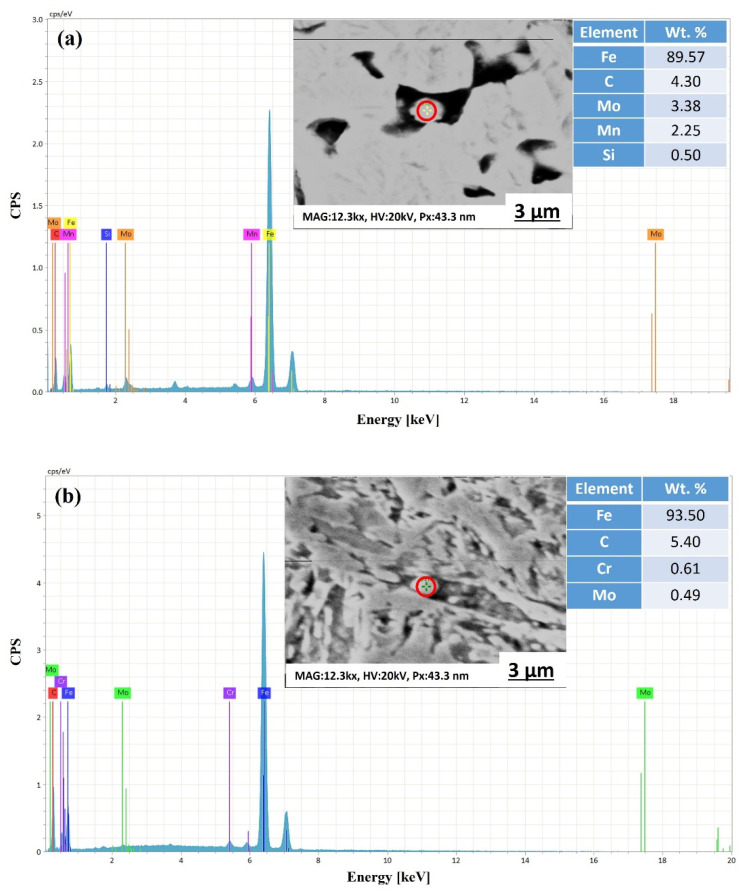
EDS spectra of LT samples at different locations, i.e., (**a**,**b**). Precipitates were measured using point scanning.

**Figure 4 materials-15-02204-f004:**
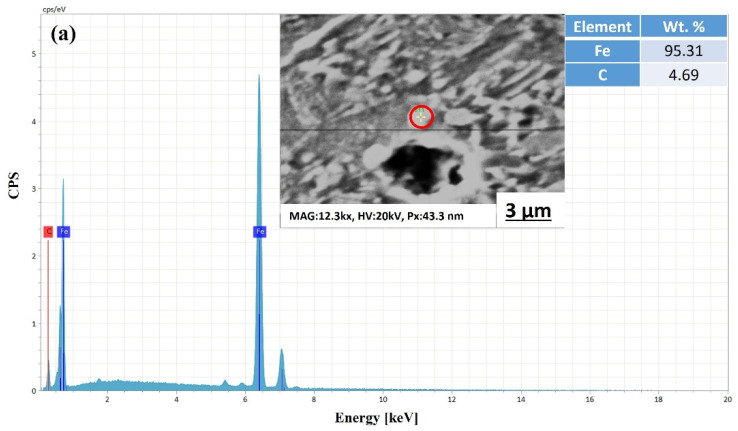
EDS spectra of HT samples at different locations, i.e., (**a**,**b**). Precipitates were measured using point scanning.

**Figure 5 materials-15-02204-f005:**
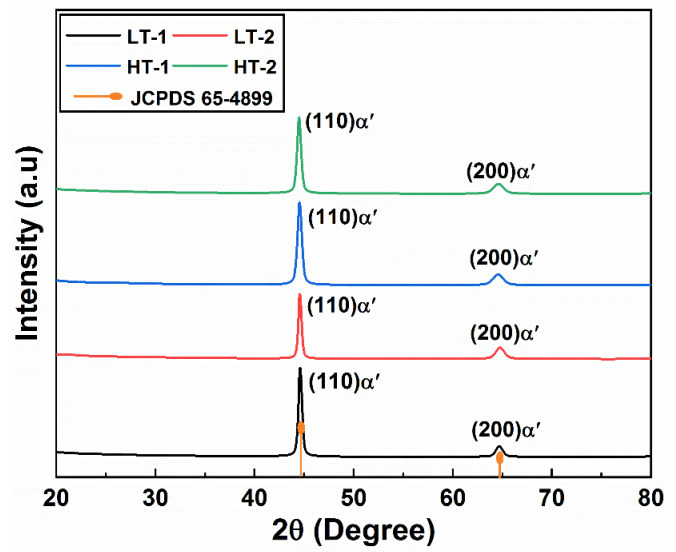
X-ray diffraction pattern of ballistic steel samples.

**Figure 6 materials-15-02204-f006:**
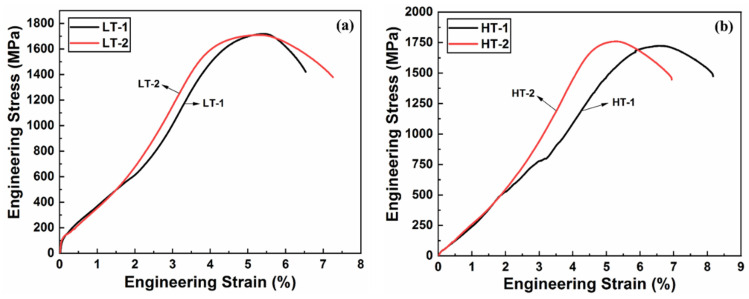
Engineering stress and strain plots of ballistic steel samples: (**a**) LT-1/LT-2 and (**b**) HT-1/HT-2.

**Figure 7 materials-15-02204-f007:**
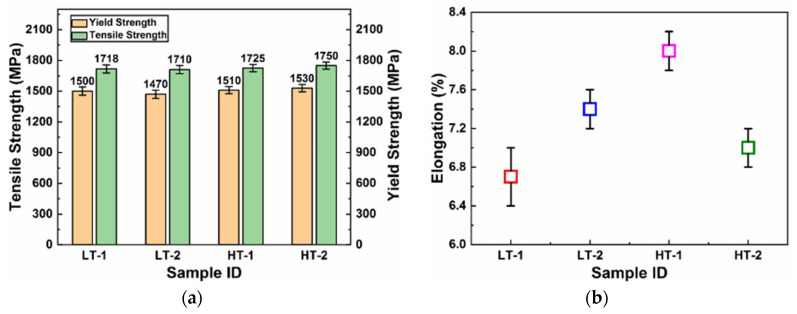
Tensile properties of ballistic steel samples: (**a**) yield and tensile strength; (**b**) percent elongation.

**Figure 8 materials-15-02204-f008:**
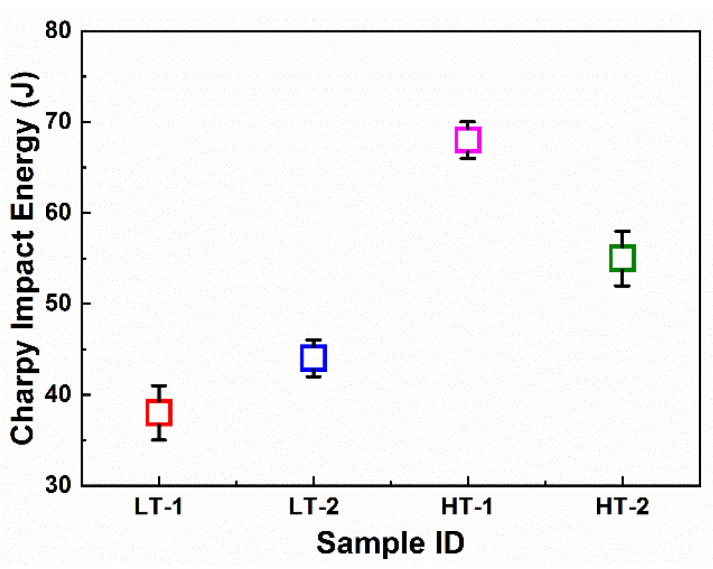
Charpy impact properties of ballistic steel samples.

**Figure 9 materials-15-02204-f009:**
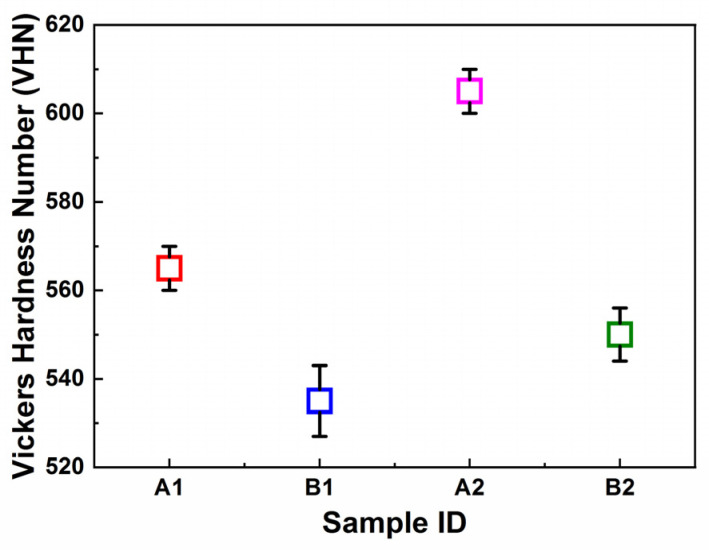
Vickers microhardness of ballistic steel samples.

**Figure 10 materials-15-02204-f010:**
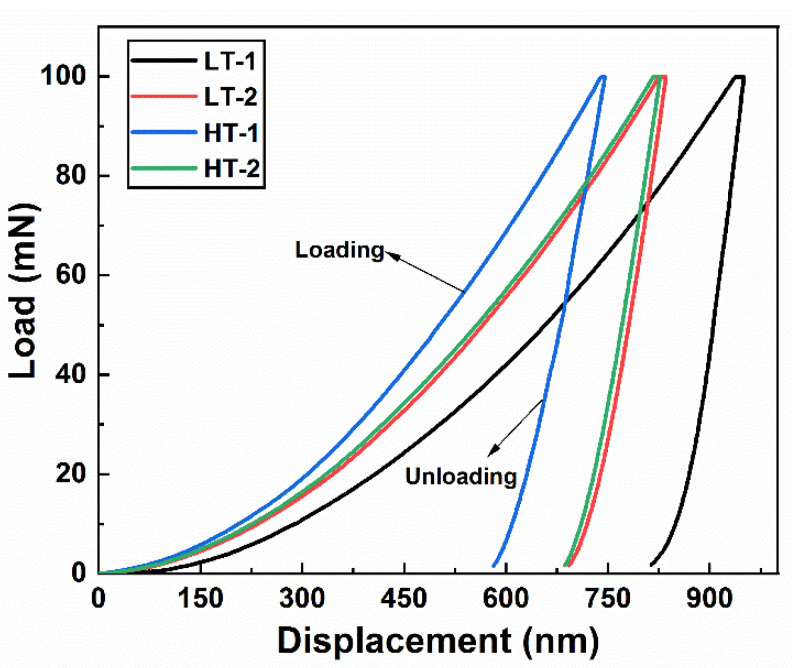
Load–displacement curves of ballistic steel samples.

**Figure 11 materials-15-02204-f011:**
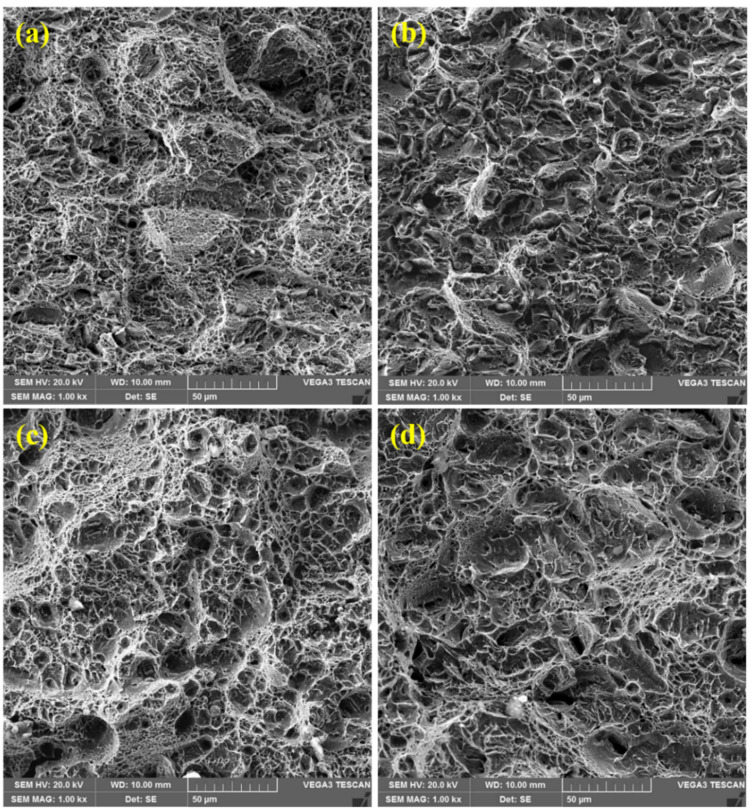
SEM fractographs of tensile samples: (**a**) LT-1; (**b**) LT-2; (**c**) HT-1; (**d**) HT-2. All of these samples exhibit ductile fractures.

**Figure 12 materials-15-02204-f012:**
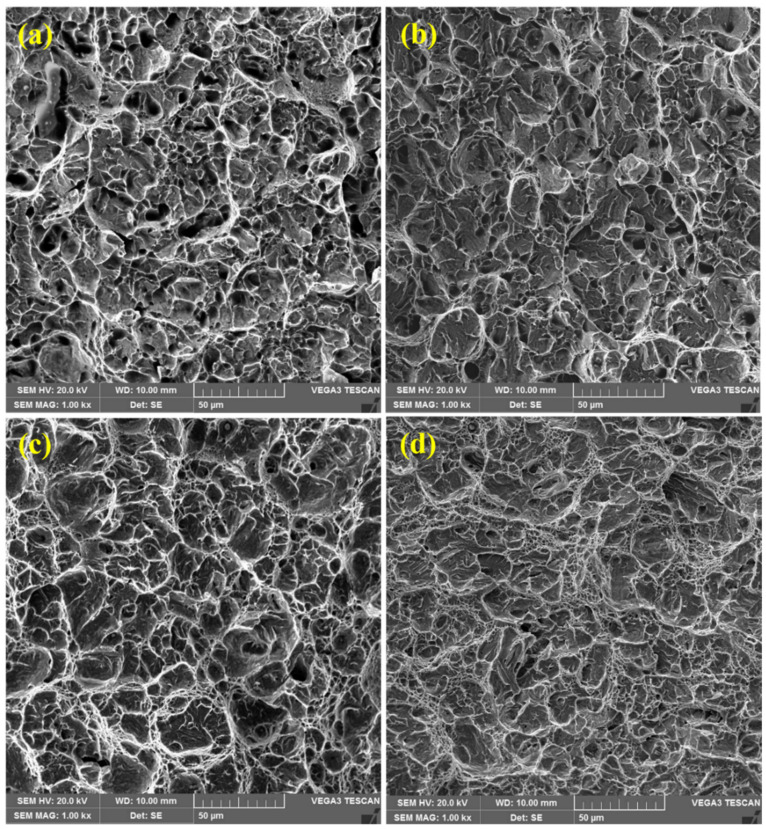
SEM fractographs of impact tested samples: (**a**) LT-1; (**b**) LT-2; (**c**) HT-1; (**d**) HT-2. Similarly to tensile counterparts, all samples exhibited ductile fractures.

**Figure 13 materials-15-02204-f013:**
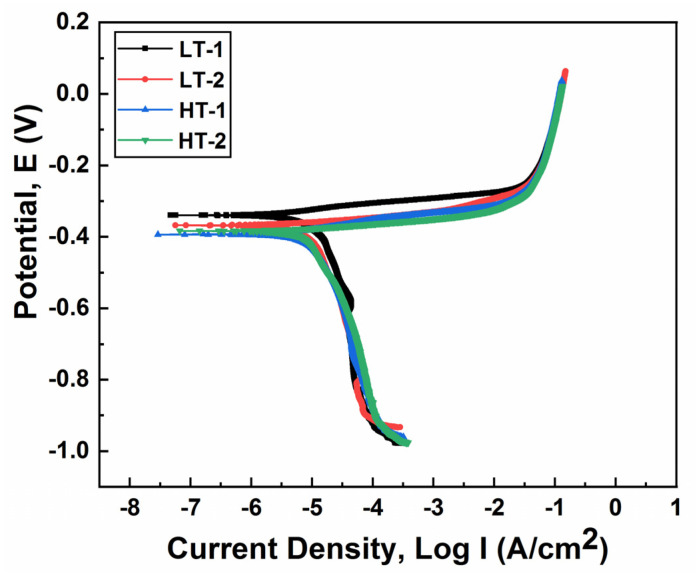
Potentiodynamic polarization curves of ballistic steel samples in 3.5 wt.% NaCl solution.

**Table 1 materials-15-02204-t001:** The materials utilized in the present study.

S.No.	Sample ID	Material Details	Thickness	Threat Level	Supplier
1	*LT-1	Ballistic Steel	7.0 mm	FB6	Pak Armoring
2	*LT-2	Ballistic Steel	6.7 mm	FB6	Streit Pakistan
3	**HT-1	Ballistic Steel	15.0 mm	FB7	Pak Armoring
4	**HT-2	Ballistic Steel	13.0 mm	FB7	Streit Pakistan

*LT = low-thickness plate, **HT = high-thickness plate.

**Table 2 materials-15-02204-t002:** The chemical composition of standard material (Armox 500T) (SSAB, Stockholm, Sweden) used for potential armoring applications.

Chemical Composition (wt. %)										
**Elements**	**C**	**Si**	**Mn**	**P**	**S**	**Cr**	**Mo**	**Ni**	**B**	**Fe**
Standard Limit	0.32 max	0.50 max	1.30 max	0.010 max	0.005 max	1.00 max	0.70 max	1.80 max	0.005 max	Balance

**Table 3 materials-15-02204-t003:** The mechanical properties of standard material (Armox 500T) used for potential armoring applications.

Material	Mechanical Properties				
**Armox 500T**	**Yield Strength (MPa)**	**Tensile Strength (MPa)**	**Elongation (%)**	**Vickers Hardness Number (VHN)**	**Impact Toughness (J)**
	1250–1450	1450–1750	7–10	530–620	Min 25

**Table 4 materials-15-02204-t004:** Chemical composition of ballistic steel samples.

Sample ID	Chemical Composition (wt. %)									
	C	Mn	Si	P	S	Cr	Ni	Mo	B	Fe
LT-1	0.26	0.90	0.30	0.006	0.001	0.90	0.96	0.26	0.003	Balance
LT-2	0.29	1.35	0.55	0.009	0.003	0.70	0.44	0.27	0.002	Balance
HT-1	0.30	1.00	0.38	0.005	0.004	0.60	0.38	0.42	0.004	Balance
HT-2	0.30	1.30	0.52	0.007	0.002	0.60	0.40	0.34	0.003	Balance

**Table 5 materials-15-02204-t005:** Mechanical properties of ballistic steel samples.

Sample ID	Yield Strength (MPa)	Tensile Strength (MPa)	Elongation (%)	Charpy Impact Energy (J)	Vickers Hardness Number (VHN)
LT-1	1500	1718	6.7	38	535
LT-2	1470	1710	7.4	44	560
HT-1	1510	1725	8.0	68	590
HT-2	1530	1750	7.0	55	540

**Table 6 materials-15-02204-t006:** Micro-mechanical properties of ballistic steel samples.

Sample ID	Hardness (H) (GPa)	Elastic Modulus (E) (GPa)	Stiffness (S) (mN/nm)
LT-1	6.80	265.87	0.9880
LT-2	7.95	244.26	0.8199
HT-1	8.20	240.94	0.7908
HT-2	7.42	252.75	0.9572

**Table 7 materials-15-02204-t007:** Corrosion parameters of ballistic steel samples obtained from Tafel curves.

Sample ID	I_corr_ (A/cm^2^)	E_corr_ (V)	Corrosion Rate (mm/Year)
LT-1	3.60 × 10^−6^	−0.340	0.0418
LT-2	4.10 × 10^−6^	−0.368	0.0476
HT-1	5.30 ×10^−6^	−0.394	0.0615
HT-2	5.10 × 10^−6^	−0.384	0.0592

## Data Availability

Not Applicable.

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
