# Peer review of "Characterization of Microstructure, Phase Composition, and Mechanical Behavior of Ballistic Steels"

_materials, 2022, doi:10.3390/ma15062204_

Round 1

Reviewer 1 Report

An article discovering new features of materials being in use. Good motivation part and good research methodology. Please, provide the comparison of measured material characteristics with those provided by the manufacturer: hardness, composition. Also, the heat treatment mode of the studied materials should be provided. In the discussion, there is no statement or a solution. Please, add a paragraph with an explanation of what properties are good, and what properties should be improved.

Also, please make some corrections of English and paper style/presentation

Author Response

Thank you!

Reviewer 2 Report

The research ”Nanoindentation and Microstructural Characterization of Ballistic Steels” is suitable for publication in the Materials journal with very short modifications.

The authors have performed very good and average original research regarding ballistic steels with a very good variety of analyses. The content presents a lot of results from the point of view of microstructure, mechanical and corrosion properties. The authors describe the results very well with proper explanations.

Some remarks:

  • please add a correspondence of the steels with a standard;
  • rewrite the conclusions with a recommendation regarding which is the better steel for usage in the military industry;
  • please specify the software that was used for the preparation of the graphs.

Author Response

Thank you!

Reviewer 3 Report

The authors study thin (6.7–7.0 mm) and thick (13.0–15.0 mm) samples of the existing state-of-art ballistic steels using tensile, Charpy, micro-vickers, nanoindentation, XRD, SEM-EDS, and corrosion tests for their characterization. During the SEM-EDS study, they found ε-carbide precipitates in thin samples and both ε-carbides and M-carbides (Molybdenum carbides) in thick samples as well as revealed differences in hardness, elastic modulus, and stiffness. For both variants, ductile fracture mechanism was observed. Also, peculiarities of corrosion behavior in 3.5 wt.% NaCl solution were substantiated. The paper is interesting, but some deficiencies need to be corrected to make it acceptable for publication.

(1) Lines 135–136: a therm “micro-vickers hardness testing” is incorrect. It is suggested “Vickers microhardness testing” to be used.

(2) Figs.5 and 6: the magnification scales on microstructure images cannot be recognized.

(3) It is preferable to use SI units for microhardness measurements. If the authors still use a VHN unit, they are suggested to give “the Vickers hardness number (VHN)” in the beginning to explain this abbreviation.

(4) The authors analyze the effect of sample thickness on tensile properties (lines 213–221), impact properties (lines 231–238), microhardness (lines 242–248), and elastic modulus and stiffness of materials (lines 252–275). The explanation looks like “This is due to the homogeneous distribution of lath martensite and carbides precipitation” (lines 272–273), but this is quite poor explanation. The authors are suggested to analyze all these tendencies in relation to microstructural changes caused, in particular, by deformation pretreatment as well as other factors. This may be done as a final discussion. The authors may use after line 301 the following text (or something like this): “3.5.3. Concluding remarks on the microstructure related mechanical behavior of tested samples

The morphology of fracture surfaces of samples after both the tensile and impact tests show a relation between mechanical behavior of the samples and microstructure of corresponding materials. As mentioned above, all microstructures were characterized by lath martensite morphology. However, a great degree of homogeneity of tempered martensite was seen in HT samples compared to that of LT counterparts. In LT samples (Fig. 3a, b and Fig. 4a, b), quite distinct packets of martensite laths can be seen. The average size of the packets is about 7–10 μm. For HT-2 sample, packets of martensite laths are about the same size but their contours are blurred (Fig. 3d). In contrast to this, a homogeneous lath martensite microstructure is observed for HT-1 sample (Fig. 3c and Fig. 4c). As can be seen in Fig.13c and, especially, in Fig.14c, fracture surface of HT-1 sample contains plastic elongation crests surrounding a set of dimples. Such crests are of a greater size (about 50–100 μm, see Fig.14c) and more distinct than those of other variants. That is why at similar levels of tensile strength, a greater elongation is achieved for HT-1 sample (see Table 3). Such a ductile fracture micromechanism also explains the fact of increased toughness (impact resistance) due to enhanced ductility of material (Table 3).”

In my opinion, without an explanation like this, the manuscript looks rather as a simple scientific report, not as a scientific article.

(5) The title of the manuscript should be optimized because the word “Nanoindentation” does not reflect all the methods used in the work. It may be as follows: “Study of Microstructure, Phase Composition, and Mechanical Behavior of Ballistic Steels” or “Characterization of Microstructure, Phase Composition, and Mechanical Behavior of Ballistic Steels” or something like this.

Author Response

Thank you!

Round 2

Reviewer 3 Report

Please remove quotation marks (inverted commas) in the end of the section 3.5.3.
All the reviewer's comments have been taken into account. Now the manuscript may be published.
Glory to Ukraine! Glory to the heroes!